# Impact of Optimal Silane Concentration on the Rheological Properties and 3D Printing Performance of Al_2_O_3_-Acrylate Composite Slurries

**DOI:** 10.3390/ma17225541

**Published:** 2024-11-13

**Authors:** Kook-Hyun Ryu, Ung-Soo Kim, Jin-Ho Kim, Jung-Hoon Choi, Kyu-Sung Han

**Affiliations:** Korea Institute of Ceramic Engineering & Technology, Icheon 17303, Gyeonggi-do, Republic of Korea; cj2303@naver.com (K.-H.R.); ukim@kicet.re.kr (U.-S.K.); jino.kim@kicet.re.kr (J.-H.K.); eunu@kicet.re.kr (J.-H.C.)

**Keywords:** silane coupling agent, Al_2_O_3_ slurry, rheology, photocuring, internal structure analysis

## Abstract

In this study, 3-trimethoxy-silylpropane-1-thiol (MPTMS) was used as a surface modifier for Al_2_O_3_ powder to systematically analyze the effects of MPTMS concentration on the rheological properties, photocuring characteristics, and 3D printing performance of photocurable composite slurries. MPTMS concentration significantly influenced the rheological behavior of the slurry. Slurries containing 2 wt.% and 5 wt.% MPTMS exhibited a wide linear viscoelastic range (LVR). However, at concentrations of 10 wt.% and 20 wt.%, the LVR range narrowed, which led to reduced dispersion stability. In dispersion stability tests, the slurry with 2 wt.% MPTMS showed the most stable dispersion, while the 5 wt.% MPTMS concentration exhibited the highest photocuring rate. In 3D printing experiments, the 5 wt.% MPTMS concentration resulted in the most stable printed structures, whereas printing failures occurred with the 2 wt.% concentration. At 10 wt.% and 20 wt.%, internal cracking was observed, leading to structural defects. In conclusion, MPTMS forms silane bonds on the Al_2_O_3_ surface, significantly impacting the stability, rheological properties, and printing quality of Al_2_O_3_-acrylate composite slurries. An MPTMS concentration of 5 wt.% was found to be optimal, contributing to the formation of stable and robust structures.

## 1. Introduction

Three-dimensional printing technology has established itself as an innovative manufacturing method across various ceramic industries, and with the increasing demand for high-precision structures, UV-curable slurry-based 3D printing technology has garnered attention [1,2,3]. However, to achieve high-quality 3D printing, it is essential to optimize the slurry’s rheological properties, dispersion stability, and photopolymerization characteristics. Alumina (Al_2_O_3_), a ceramic material with high thermal stability, chemical stability, and strength, is widely used in various technical applications. However, due to the high agglomeration and specific surface area of Al_2_O_3_ powder, maintaining uniform dispersion in the slurry is challenging [4,5]. To address this issue, surface modification techniques such as silanization are employed, where a silane coupling agent (SCA) is adsorbed onto the powder surface to enhance dispersion stability within the slurry [6,7,8,9].

SCAs are compounds whose molecules contain functional groups that bond with both organic and inorganic materials [10]. The organic functional group (X) consists of alkyl, vinyl, methacrylic, and other groups, helping the silane to be compatible with organic materials (monomers). The silicon functional group (OR) typically consists of alkoxy groups like methoxy and ethoxy, facilitating hydrolysis reactions with methanol and ethanol [11,12]. An SCA acts as an intermediary, bonding organic materials to inorganic materials. This characteristic makes SCAs useful for improving the mechanical strength of composite materials, enhancing adhesion, and modifying resins and surfaces.

Many studies have been conducted to improve the properties of 3D printing processes and photocurable structures using various ceramic powders modified with SCAs [6,7,8,9]. The powders modified with SCAs exhibit excellent dispersibility, improving the uniformity and fluidity of the slurry, which in turn enhances the mechanical properties of the printed ceramic polymer composites. However, most studies have focused on comparing and evaluating different types of SCAs or conducting experiments with lower concentrations of additives for practical manufacturing applications. Some studies have concentrated on analyzing the chemical bonding formed when powders modified with SCAs are mixed with monomers or oligomers at each stage of the experiment [13,14,15]. While research has been reported on a wide range of SCA concentrations to increase the solid content of the slurry, these studies have mostly focused on changes in slurry viscosity, without analyzing the effects on photocuring characteristics [16]. Some studies have reported changes in material properties after curing, but systematic research on the changes in material characteristics at each process stage and the correlation between those characteristics and the properties after curing is still lacking [17,18,19].

In this study, 3-trimethoxy-silylpropane-1-thiol (MPTMS) was used as a surface modifier for Al_2_O_3_ powder to systematically analyze the effect of MPTMS concentration on the rheological properties, photocuring characteristics, and 3D printing performance of the slurry. MPTMS plays a crucial role in controlling particle interactions by forming silane bonds on the Al_2_O_3_ surface, thereby adjusting the slurry’s stability and curing rate. The primary objective was to identify the optimal MPTMS concentration that maximizes the physical properties of the Al_2_O_3_ slurry and enhances 3D printing performance. This research will provide fundamental data crucial for the production of high-quality ceramic components.

## 2. Materials and Methods

### 2.1. Materials

In this experiment, Al_2_O_3_ powder was purchased from Sumitomo, Japan (AES-11H). Methyl alcohol (CH_3_OH, extra pure, Daejung, Siheung-si, Republic of Korea) was used as a solvent. The SCA used was 3-trimethoxy-silylpropane-1-thiol (MPTMS) with an average molecular weight of Mw = 196.34 g/mol (KBM 503, Shin-Etsu Chemicals Co., Ltd., Tokyo, Japan). The monomer used was trimethylolpropane triacrylate (TMPTA) with an average molecular weight of Mw = 296.32 g/mol. Phenylbis (2,4,6-trimethyl benzoyl) phosphine oxide (Irgacure 819) was purchased from Sigma Aldrich (Burlington, MA, USA) and used as a photoinitiator.

### 2.2. Adsorption of MPTMS on Al_2_O_3_ Powder

Methyl alcohol and Al_2_O_3_ were mixed in a 6:4 ratio and ball-milled for 24 h. MPTMS was then added at 2, 5, 10, and 20 wt.% relative to Al_2_O_3_, followed by heating at 50 °C for 24 h using a heating mantle to facilitate hydrolysis and condensation reactions, allowing MPTMS to bond to the Al_2_O_3_ surface [16]. To remove the solvent from the slurry, it was centrifuged at 5000 rpm for 30 min (1696R, Rabogene, Gimpo-si, Republic of Korea), and the precipitate was rinsed three times with methanol and acetone. The cleaned precipitate was then dried in a vacuum oven (SOV-20, Daihan Scientific, Wonju-si, Republic of Korea) set at 60 °C for 5 h.

The amount of MPTMS adsorbed on the Al_2_O_3_ surface was measured using thermogravimetric analysis (TGA; STA7200RV, Hitachi, Tokyo, Japan). The weight of the thermally decomposed material was measured three times between 25 °C and 800 °C with a heating rate of 10 °C/min, and the average value was reported. To confirm the functional groups of the MPTMS-adsorbed Al_2_O_3_, Fourier-transform infrared spectroscopy (FT-IR; Nicolet 6700, Thermo Fisher Scientific, Bremen, Germany) was conducted on the dried particles using the transmittance mode in the range of 600–4000 cm^−1^ on an ATR crystal.

### 2.3. Preparation and Characterization of Al_2_O_3_ Acrylate Composite Slurry

The MPTMS-modified Al_2_O_3_ slurry prepared as described above was mixed with TMPTA and allowed to react for 12 h at room temperature to induce chain polymerization between MPTMS and TMPTA [20]. The solvent was then evaporated at 50 °C using a vacuum evaporator (N-1200A, EYELA, Tokyo, Japan) to obtain the solvent-free slurry. Afterward, 1 wt.% of the photoinitiator was mixed for 24 h to prepare the final solvent-free Al_2_O_3_-acrylate composite slurry, which had a solid content of 60 wt.%.

A rheometer (HAAKE MARS III, Thermo Fisher Scientific Inc., Bremen, Germany) was used to analyze the rheological behavior of the slurry. The experiment was conducted at a constant temperature (25 °C) using a cup-and-bob (Coaxial Cylinders, Φ = 16 mm) measuring system. The cup-and-bob configuration is beneficial for low-viscosity fluids because it increases the total surface area and, consequently, the viscous drag on the rotating inner cylinder, thereby enhancing the accuracy of the measurements [21,22]. Shear stress was measured under the steady-state mode with a shear rate ranging from 10^−4^ to 10^3^ s^−1^. To evaluate the deformation behavior and viscoelastic properties of the slurry, amplitude oscillation sweep measurements were conducted. In the linear viscoelastic range (LVR) determined from the amplitude oscillation sweep, a frequency oscillation sweep (angular frequency = 0.1–100 rad/s) was performed.

To analyze the dispersion stability, a sedimentation test was conducted using a Turbiscan LAB stability analyzer (Formulaction SA, Toulouse, France). Cylindrical glass cells were used for the measurement, and an 880 nm near-infrared wavelength was employed as the light source. The backscattering (%) profile was used to graph destabilization phenomena, which included particle migration (creaming/sedimentation) and particle size variation (flocculation/coalescence).

The photocuring behavior of the slurry was analyzed using a photo-DSC (DSC 204 F1 Phoenix, Netzsch, Selb, Germany). The sample was placed in an aluminum crucible with a pierced lid, and measurements were taken at the same temperature (25 °C), light source (405 nm), and light intensity (5.4 W/cm^2^), with varying exposure times (1.2, 2.4, and 4.8 s). The photocuring conversion rate was calculated based on the curing enthalpy (ΔH) measured by the photo-DSC [23].

### 2.4. Three-Dimensional Printing and Internal Structure Analysis

To verify the 3D printing behavior of the solvent-free Al_2_O_3_-acrylate composite slurry, disk-shaped samples (diameter = 10 mm, thickness = 2 mm) were printed using a DLP 3D printer (IM96, Carima, Seoul, Republic of Korea). The basic/primary exposure times were varied as 1.2/5.0, 2.4/10, and 4.8/20 s during printing. The basic exposure time refers to the time taken for the slurry to adhere to the build platform during the first 10 layers of the total 204-layer printing process. Primary exposure time refers to the exposure time for all the remaining layers except for the basic exposure time.

The internal structure of the Al_2_O_3_-acrylate composite printed via 3D printing was examined using a Micro Focus X-ray CT system (InspeXio SMX-225CT, Shimadzu, Shenzhen, China). The captured images were reconstructed into 3D images using the myVGL ver. 3.2 program, and the internal structures of the Al_2_O_3_ composites were compared based on MPTMS content.

### 2.5. Statistical Analysis

The significance test between each treatment group for the experimental results was conducted using statistical analysis with Minitab Version 14 (Minitab, Inc., State College, PA, USA) and verified at a significance level of *p* < 0.05. The change in adsorption amount according to the SCA addition levels was analyzed using a one-way ANOVA to determine if the results were statistically significant. TSI values of the slurries with time for different SCA concentrations were analyzed using regression analysis.

## 3. Results and Discussion

### 3.1. Concentration-Dependent Adsorption of SCA on Al_2_O_3_

The weight change of MPTMS-modified Al_2_O_3_ as a function of temperature is summarized in Figure 1.

The weight loss occurred in two stages: the first stage was observed in the temperature range of 300–380 °C, and the second stage appeared between 400–500 °C. The first stage was attributed to the thermal decomposition of the organic solvents used for cleaning, with the amount of weight loss falling between 0.07 and 0.09 wt.%. The second stage reflected the thermal decomposition of MPTMS adsorbed on the Al_2_O_3_ surface. The weight loss in this second stage, corresponding to the decomposition of the SCA, was 0, 0.18, 0.11, 0.14, and 0.25 wt.% as the SCA content increased.

Figure 2 shows the variation in the amount of the SCA adsorbed as a function of SCA concentration. The change in the adsorption amount according to the SCA addition levels was analyzed using an ANOVA (Minitab, Inc., USA) to determine if the results were statistically significant. The analysis yielded *p* < 0.001, indicating that the change in the adsorption amount due to the varying SCA addition levels was meaningful.

The adsorption amount increased up to an SCA concentration of 2 wt.%, decreased at 5 wt.%, and then increased again at concentrations above 10 wt.%. Interestingly, at an SCA concentration of 5 wt.%, the adsorption amount was lower than that at 2 wt.%, despite the higher SCA concentration. J. Quinton et al. reported an oscillatory adsorption behavior of SCA depending on the SCA concentration and adsorption time. They explained this behavior through a model where organosilane molecules initially bind strongly to the oxide surface, but after a certain period, they desorb and then re-adsorb onto the surface [24,25]. As a result, the adsorption amount does not increase linearly with the SCA concentration or adsorption time but follows an oscillatory adsorption pattern, showing a cycle of increase–decrease–increase.

In this context, the experimental results of SCA adsorption as a function of concentration suggest that at 2 wt.% of the SCA, only the initial adsorption reaction occurred, while at 5 wt.% of the SCA, both adsorption and desorption reactions happened simultaneously, leading to re-adsorption. As the SCA concentration increased to 10 wt.% or more, the re-adsorption became more pronounced, resulting in a further increase in the adsorption amount. Consequently, the amount of the SCA adsorbed on the Al_2_O_3_ surface did not increase linearly with SCA concentration but followed a trend of 20 > 2 > 10 > 5 > 0 wt.%, showing this oscillatory adsorption behavior.

### 3.2. FT-IR Analysis of SCA Bonding on Al_2_O_3_ Surface

To confirm the bonding state of SCA onto the surface of Al_2_O_3_, the FT-IR measurement results are summarized in Figure 3.

The FT-IR spectrum of Al_2_O_3_ powder without the SCA did not exhibit any significant absorption peaks, and most of the peaks appeared broad. In contrast, all the Al_2_O_3_ samples with the SCA (2, 5, 10, 20 wt.%) showed sharp peaks of a similar shape.

Regardless of the SCA concentration, all samples displayed peaks at 3450, 2359, 2338, and 1659 cm^−1^, which corresponded to the Al-O-H bonds onto the surface of Al_2_O_3_. The peaks at 2980 and 2885 cm^−1^ represented C-H stretching vibrations between Al_2_O_3_ and SCA, while the peak at 1045 cm^−1^ in the 1200–950 cm^−1^ region indicated the formation of Al-O-Si and Si-O-Si bonds. Since the masses of Al and Si atoms are similar, this peak appeared at a vibration frequency close to that of Al-O-Al bonds. These peaks demonstrated the bonding interaction between Al_2_O_3_ and the SCA [26].

The peak at 2980 cm^−1^ corresponded to asymmetric CH_3_ and CH_2_ stretching, while the 2885 cm^−1^ peak represented the symmetric stretching of the CH_3_ group in OCH_3_ [27]. The 1045 cm^−1^ peak indicated asymmetric and symmetric Si-O-CH_3_ stretching, suggesting that Si-O-Si bonds are formed as a result of hydrolysis and condensation reactions in the alkoxy groups in the SCA. This also implies that double bonds are generated by the acrylate group of the SCA. As such, it can be confirmed that the SCA forms the same type of bonding on the Al_2_O_3_ surface, regardless of the SCA concentration.

### 3.3. Rheological Behavior of Photocurable Al_2_O_3_ Slurries

To investigate the rheological behavior of the photocurable slurry prepared by adding TMPTA to MPTMS-modified Al_2_O_3_ slurry, viscoelastic analyses were conducted using a rheometer. Figure 4 presents the results of the amplitude oscillation sweep measurements.

All slurries exhibited elastic deformation behavior within the LVR with (G’ > G”). The LVR range for the slurries containing 2 and 5 wt.% of the SCA was 10^−3^~1 Pa, while the slurries with a higher SCA content, 10 and 20 wt.%, showed a relatively shorter LVR range of 10^−3^~10^−1^ Pa. As the SCA content increased, the LVR range decreased, indicating a weakening in the slurry’s resistance to deformation. In other words, the slurries with 2 and 5 wt.% of the SCA had higher resistance to deformation compared to those with 10 and 20 wt.% of the SCA.

Using the G’ and G” obtained from the oscillation sweep measurements, the loss factor (tan δ) was calculated. This loss factor, also known as the sol/gel transition point, provides insight into the ratio between the two components of viscoelastic behavior. The values of the loss factor are presented over a frequency range of 0.1–10 Hz (Figure 5a).

The results were consistent with the previous viscoelastic behavior experiment, with all samples showing tan δ = G”/G’ < 1, indicating that the slurry remained in a gel state without any changes in properties during measurement. As the SCA content decreased, the tan δ values also decreased, meaning that the lower the SCA content, the greater the proportion of elasticity in the viscoelastic properties.

Figure 5b shows the changes in the shear stress of the slurries with varying shear rates. The shear stress increased with an increase in the shear rate of all suspensions, confirming typical pseudoplastic behavior [3,28,29]. At a constant shear rate, the shear stress decreased as the SCA content increased up to 10 wt.%. At a very low shear rate, the shear stress values showed little difference. However, at a high shear rate, the change in shear stress according to the SCA content was significantly more pronounced.

The data of the shear stress at different shear rates were fitted to the Herschel–Bulkley model to determine the three Herschel–Bulkley parameters [30,31]. Table 1 presents the results of the experimental data for the Herschel–Bulkley model fitting and its statistical parameters. The yield stress (τ_0_) of the slurry significantly decreased with the increase in SCA content up to 10 wt.% and showed relatively minor changes with further additions. This indicated that the stress required for fluid flow decreased with increasing SCA content. The consistency index (k) showed a similar trend to yield stress, also decreasing as the SCA content increased, indicating a reduction in the slurry’s viscosity. Although all flow index (n) values were below 1, showing that all slurries exhibited shear-thinning behavior, no specific correlation was observed with the SCA content.

The final column of the table shows the yield stress (τ_sr_) analyzed using the stress ramp test. When comparing with the fitting parameters obtained using the Herschel–Bulkley model, we observed that the trend of decreasing yield stress with an increase in SCA content was consistent. However, significant differences were noted between the fitting values and the actual values.

### 3.4. Dispersibility and Sedimentation Behavior of Photocurable Al_2_O_3_ Slurries

The dispersibility of the photocurable Al_2_O_3_ slurry with varying SCA content is shown in Figure 6.

The slurry containing 2 wt.% of the SCA exhibited an approximately 20% increase in backscattering (%) in the upper layer. This indicated the occurrence of Rayleigh diffusion, which happens when the particle size is small. As the spacing between small particles decreases, the probability of light scattering with particles increases, leading to an increase in backscattering (%) values [32,33,34]. Changes in the middle and upper layers showed relatively thin sedimentation and flocculation layers, indicating that the small particle size and uniform distribution prevented significant sedimentation, resulting in a smaller inter-particle distance and thus an increase in backscattering values.

The slurries with 5, 10, and 20 wt.% of the SCA exhibited a decrease in backscattering (%) in the upper layer due to Mie diffusion [32,33,34]. The slurry with 5 wt.% of the SCA showed the widest and deepest clarification layer and an approximately 80% reduction in backscattering (%). The slurries with 10 wt.% and 20 wt.% of the SCA showed backscattering (%) reductions of about 60% and 40%, respectively. The bimodal decrease pattern observed in all three slurries suggested the formation of aggregates. The most significant reduction observed in the 5 wt.% SCA slurry indicated the highest occurrence of aggregates. The formation of these aggregates increased the average inter-particle distance in the slurry, resulting in fewer scattering events and a decrease in backscattering values.

The Turbiscan Stability Index (TSI) of the slurries with varying SCA content was measured and is summarized in Figure 7. The relationship between time and TSI values was confirmed through regression analysis (Minitab, Inc., USA). The *p*-values for all slurries were found to be below 0.05.

After 24 h, the TSI values of the slurries with 2, 5, 10, and 20 wt.% of the SCA were approximately 1.0, 16.0, 4.0, and 2.0, respectively. The TSI values followed the trend of 5 > 10 > 20 > 2 wt.%. The TSI value for the 5 wt.% SCA slurry increased sharply after 4 h, whereas the slurries with 2, 10, and 20 wt.% of the SCA showed a gradual increase over 24 h. The sudden increase in TSI indicated the occurrence of particle aggregation and sedimentation.

### 3.5. Photocuring Behavior of Al_2_O_3_ Slurry

The photocuring behavior of the photocurable Al_2_O_3_ slurry depending on the SCA content is presented in Figure 8 and Table 2.

As the exposure time increased at the same SCA content, the maximum heat flow (ΔQ_max_), curing enthalpy (ΔH), and conversion (%) generally increased. However, the maximum values were observed when the specimen with 5 wt.% of the SCA was exposed for 4.8 s. The 5 wt.% SCA slurry was relatively agglomerated, which extended the mean free path of the photons. Therefore, as the depth of cure increased during the photopolymerization reaction, the highest values of the maximum heat flow (ΔQ_max_), curing enthalpy (ΔH), and conversion (%) were obtained.

The slurries with 2 wt.% and 20 wt.% of the SCA exhibited lower photocuring rates. The analysis of the slurries revealed that the well-dispersed Al_2_O_3_ particles caused scattering during the photopolymerization reaction, reducing the depth of cure. Consequently, these slurries showed relatively lower maximum heat flow (ΔQ_max_) and curing enthalpy (ΔH) values. The slurry with 10 wt.% of the SCA had a relatively longer mean free path due to the presence of larger aggregates, resulting in higher maximum heat flow (ΔQ_max_) and curing enthalpy (ΔH) values compared to the 2 wt.% slurry, but lower than those of the 20 wt.% slurry.

### 3.6. Three-Dimensional Printing Behavior and Internal Structure Analysis of Al_2_O_3_ Acrylate Composites

The printing behavior of the photocurable Al_2_O_3_ slurry depending on the SCA content is summarized in Figure 9, Figure 10, Figure 11 and Figure 12.

The slurry containing 2 wt.% of the SCA failed to adhere to the build platform under all exposure time conditions, resulting in failed fabrication. This phenomenon occurred because the photocuring reaction was inefficient, weakening the chain polymerization structure of the MPTMS-modified Al_2_O_3_ slurry and preventing the formation of the Al_2_O_3_ acrylate composite.

In contrast, the slurry containing 5 wt.% of the SCA formed a thin Al_2_O_3_ acrylate composite at a primary exposure time of 5 s. However, under all other exposure times, a normal Al_2_O_3_ acrylate composite was formed, leading to successful fabrication over a wide range. Due to the highest photocuring rate of the slurry, the MPTMS-modified Al_2_O_3_ slurry formed a strong network during the photopolymerization reaction, successfully forming the Al_2_O_3_ acrylate composite under all exposure conditions.

For the 10 wt.% SCA slurry, the Al_2_O_3_ acrylate composite was printed only at a primary exposure time of 5 s. For the 20 wt.% SCA slurry, the Al_2_O_3_ acrylate composite was formed at primary exposure times of 10 and 20 s. The increased printing area of the Al_2_O_3_ acrylate composite with the 20 wt.% SCA slurry as the exposure time increased suggests that the re-adsorption phenomenon, caused by oscillatory absorption, was more prominent.

The internal structure of the specimens fabricated by 3D printing was examined via micro-CT, as shown in Figure 13.

The Al_2_O_3_ acrylate composite with 2 wt.% of the SCA could not be analyzed due to the extremely thin thickness after failing to adhere to the build platform. The Al_2_O_3_ acrylate composite with 5 wt.% of the SCA showed a stable internal structure with no cracks and a high density. Cracks were observed in the Al_2_O_3_ acrylate composites with 10 and 20 wt.% of the SCA. The specimen with 10 wt.% of the SCA exhibited larger cracks than that with 20 wt.% of the SCA.

The internal structure of the printed Al_2_O_3_ acrylate composite was significantly influenced by the SCA adsorption isotherm and slurry dispersion behavior. The 5 wt.% SCA Al_2_O_3_ acrylate composite, with optimal dispersion, exhibited the most stable internal structure due to uniform photopolymerization during printing. The 20 wt.% SCA Al_2_O_3_ acrylate composite, with a higher adsorption amount than the 10 wt.% SCA one, formed stronger bonds between the alumina and SCA, resulting in fewer cracks and a more stable internal structure.

## 4. Conclusions

This comprehensive study highlights the critical role of SCA concentration in determining the stability, rheology, and printability of Al₂O₃-acrylate composite slurries, with the 5 wt.% SCA concentration emerging as the most optimal for achieving stable and robust printed structures.

MPTMS was successfully adsorbed onto the surface of Al₂O₃, forming Al-O-Si and Si-O-Si bonds. The highest adsorption was observed at the 20 wt.% MPTMS concentration, but due to the oscillatory adsorption behavior, the adsorption amount did not increase proportionally with the added amount.

The rheological properties of the slurry varied significantly depending on the MPTMS concentration. At 2 wt.% and 5 wt.% concentrations, the slurries exhibited a wide linear viscoelastic region (LVR) and formed a strong gel structure. In contrast, at 10 wt.% and 20 wt.% concentrations, the LVR region decreased, indicating a reduced resistance to deformation. All slurries exhibited shear-thinning behavior, and as the MPTMS concentration increased, the dispersion effect of the Al_2_O_3_ powder improved, resulting in lower slurry viscosity.

The dispersion stability experiments showed that the 2 wt.% MPTMS concentration provided the most stable dispersion with minimal sedimentation. However, at concentrations above 5 wt.%, particle aggregation and sedimentation became more pronounced.

The photocuring reaction rate varied with the MPTMS concentration, with the highest photocuring rate observed at 5 wt.%. This was attributed to the increased light transmittance due to particle aggregation. In contrast, at the 2 wt.% concentration, the increased light scattering caused by dispersed fine particles led to a decrease in the photocuring rate.

The success rate and quality of 3D printing were significantly influenced by the MPTMS concentration. The 5 wt.% MPTMS slurry formed stable, high-density structures and showed the best printing results. At the 2 wt.% concentration, the slurry did not adhere well to the substrate during printing, resulting in printing failure. At concentrations of 10 wt.% or higher, internal cracks occurred, leading to structural defects.

This study presents the technical guidance for preparing photocurable slurry required for the manufacturing of ceramic–polymer composites using a 3D printing process. Future research will aim to supplement the analysis of property changes during long-term storage of the slurry and the mechanical properties of the composite after curing in order to establish the technical indicators necessary for practical application in manufacturing settings.

## Figures and Tables

**Figure 1 materials-17-05541-f001:**
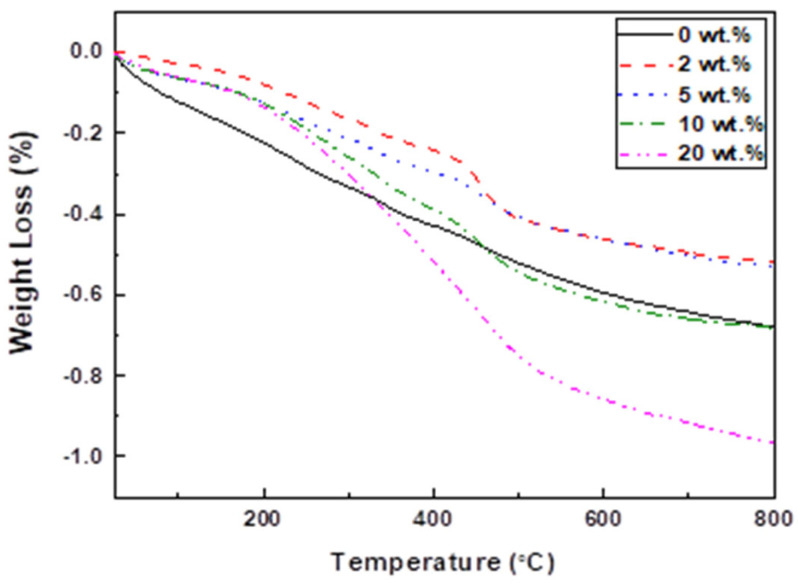
Thermogravimetric analysis of Al_2_O_3_ powders treated with various concentrations of SCA.

**Figure 2 materials-17-05541-f002:**
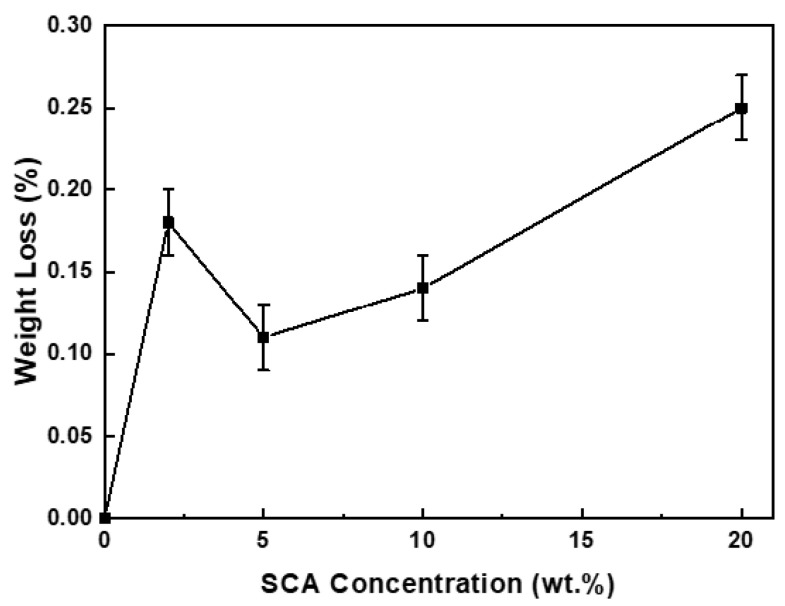
Weight loss of Al_2_O_3_ powders treated with various concentrations of SCA. The confidence level was 99.9%.

**Figure 3 materials-17-05541-f003:**
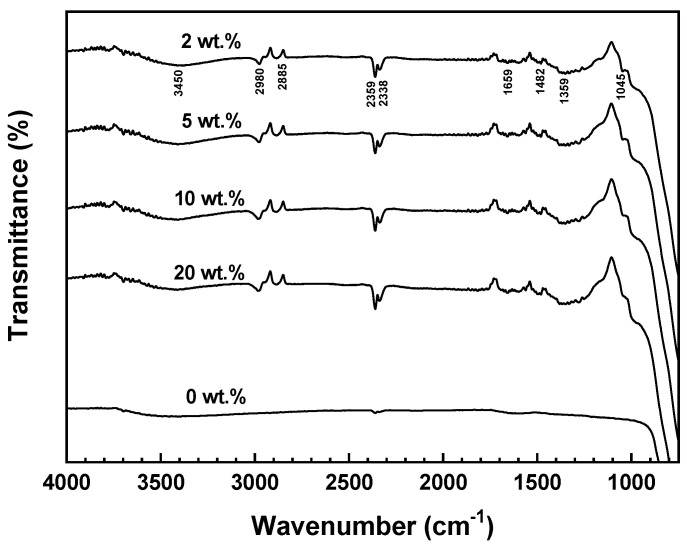
FT-IR spectra of as-received and SCA-treated Al_2_O_3_ powders.

**Figure 4 materials-17-05541-f004:**
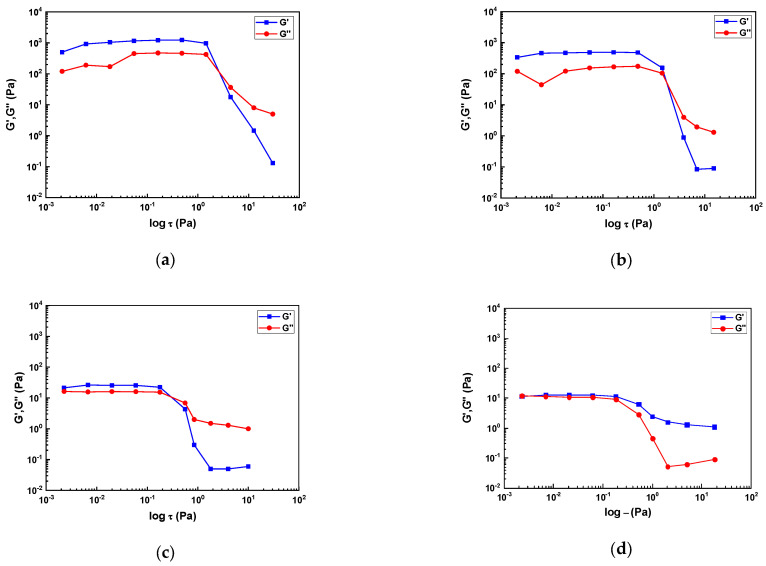
Results of the amplitude sweep test for the Al_2_O_3_ slurries with different SCA concentrations; (**a**) 2 wt.%, (**b**) 5 wt.%, (**c**) 10 wt.%, and (**d**) 20 wt.%. All the measurements were conducted at 25 °C and a frequency of 10 rad/s.

**Figure 5 materials-17-05541-f005:**
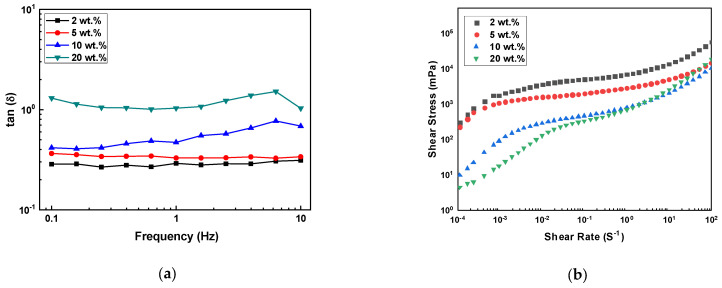
Influence of SCA concentration on the rheological properties of photocurable Al_2_O_3_ slurries; (**a**) tan δ value as a function of oscillation frequency, and (**b**) shear stress of slurries with varying shear rates.

**Figure 6 materials-17-05541-f006:**
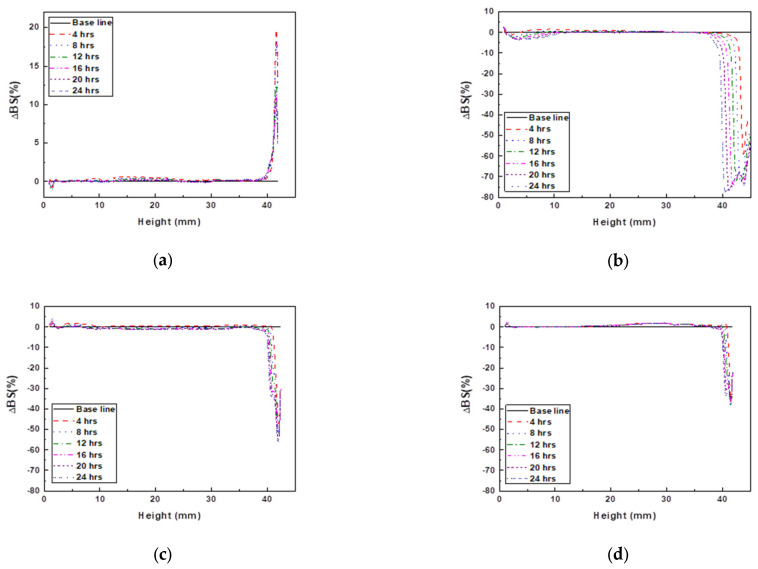
Backscattering profile of photocurable Al_2_O_3_ slurries with different SCA concentrations; (**a**) 2 wt.%, (**b**) 5 wt.%, (**c**) 10 wt.%, and (**d**) 20 wt.%.

**Figure 7 materials-17-05541-f007:**
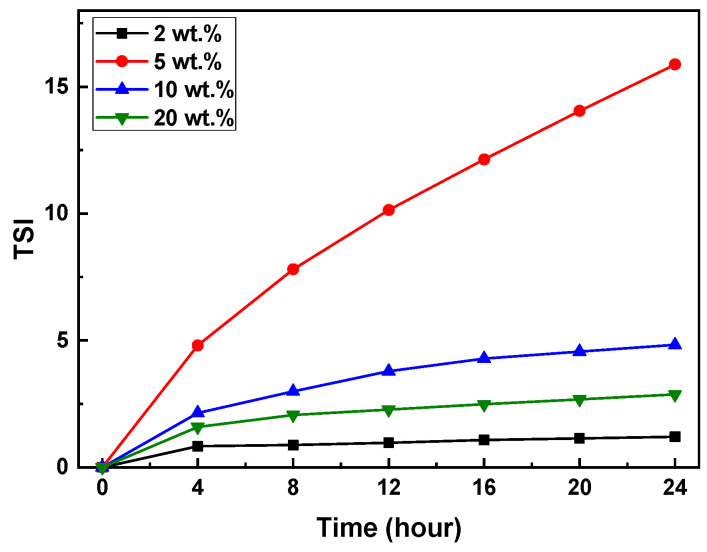
TSI curves of photocurable Al_2_O_3_ slurries with different SCA concentrations. The confidence level was 95%.

**Figure 8 materials-17-05541-f008:**
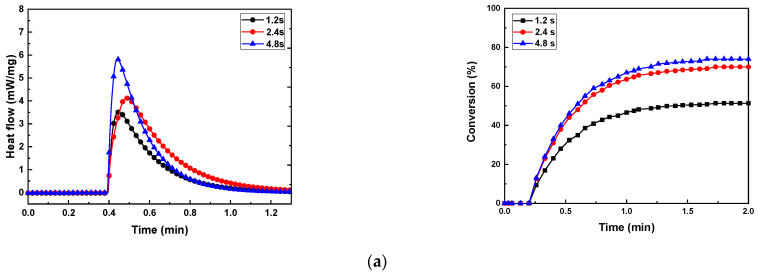
Heat flow and conversion of photocurable Al_2_O_3_ slurries with different SCA concentrations; (**a**) SCA 2 wt.%, (**b**) SCA 5 wt.%, (**c**) SCA 10 wt.%, and (**d**) SCA 20 wt.%.

**Figure 9 materials-17-05541-f009:**
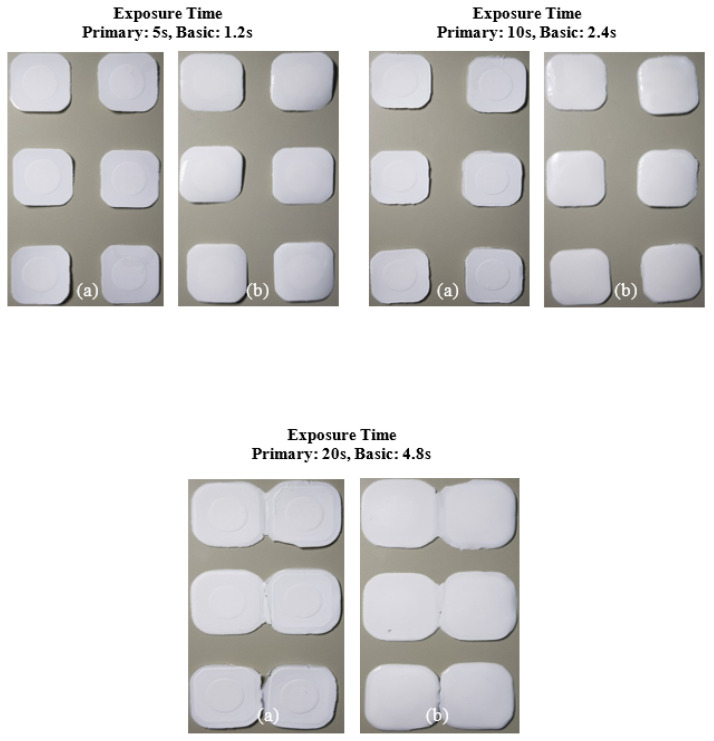
Printing results of photocurable Al_2_O_3_ slurry with 2 wt.% SCA concentration under different exposure conditions; (**a**) surface image facing build platform and (**b**) surface image facing tray.

**Figure 10 materials-17-05541-f010:**
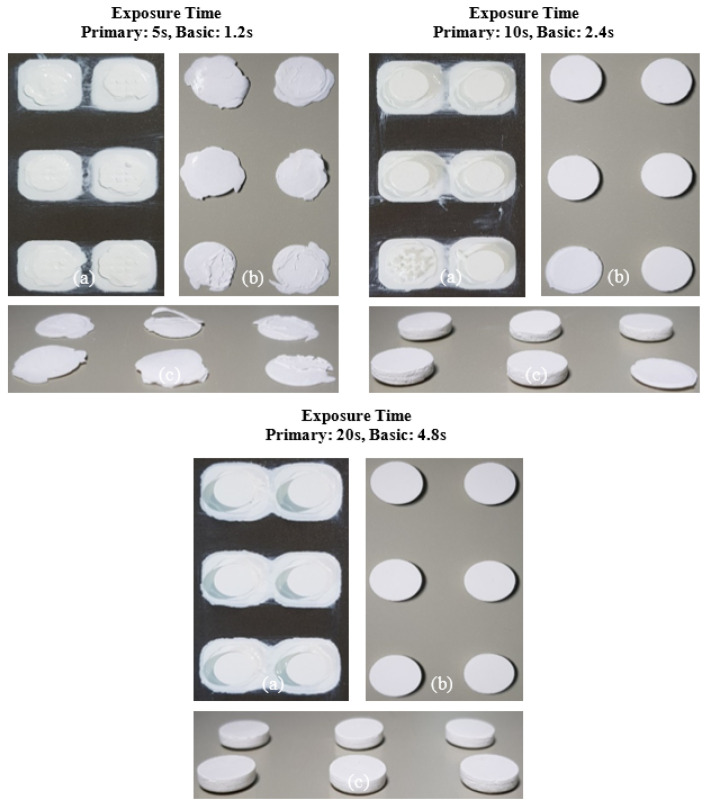
Printing results of photocurable Al_2_O_3_ slurry with 5 wt.% SCA concentration under different exposure conditions; (**a**) surface image facing build platform, (**b**) surface image facing tray, and (**c**) image of printed samples.

**Figure 11 materials-17-05541-f011:**
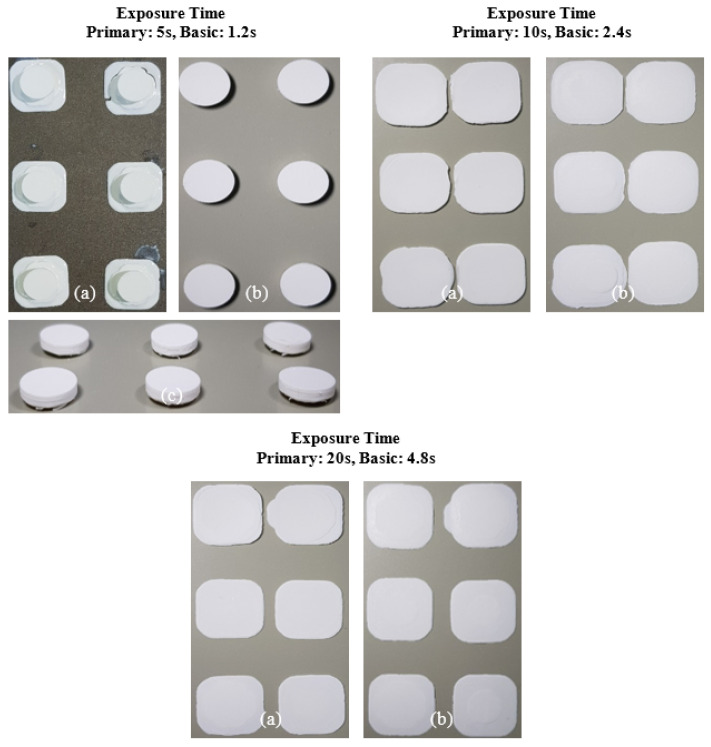
Printing results of photocurable Al_2_O_3_ slurry with 10 wt.% SCA concentration under different exposure conditions; (**a**) surface image facing build platform, (**b**) surface image facing tray, and (**c**) image of printed samples.

**Figure 12 materials-17-05541-f012:**
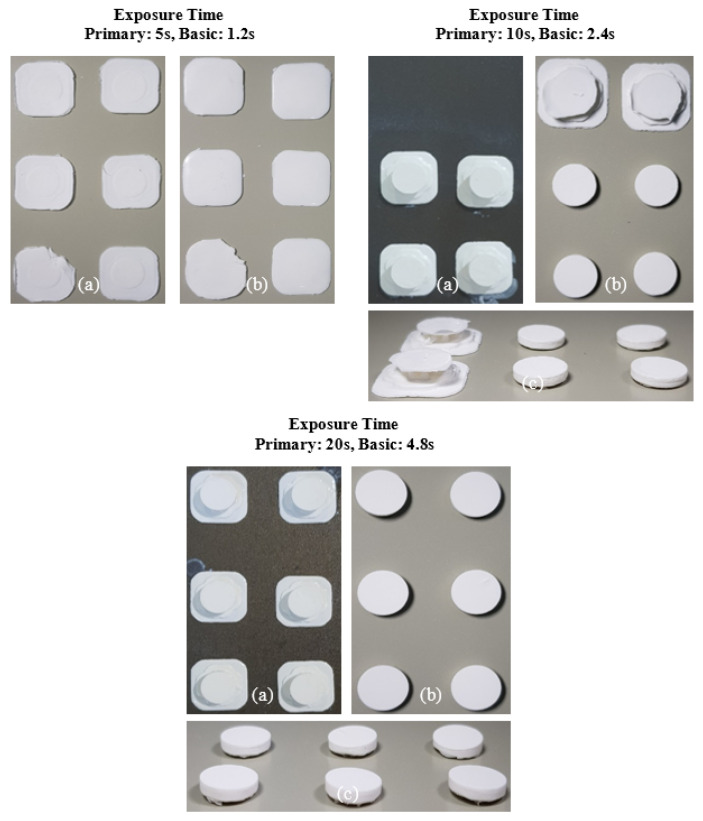
Printing results of photocurable Al_2_O_3_ slurry with 20 wt.% SCA concentration under different exposure conditions; (**a**) surface image facing build platform, (**b**) surface image facing tray, and (**c**) image of printed samples.

**Figure 13 materials-17-05541-f013:**
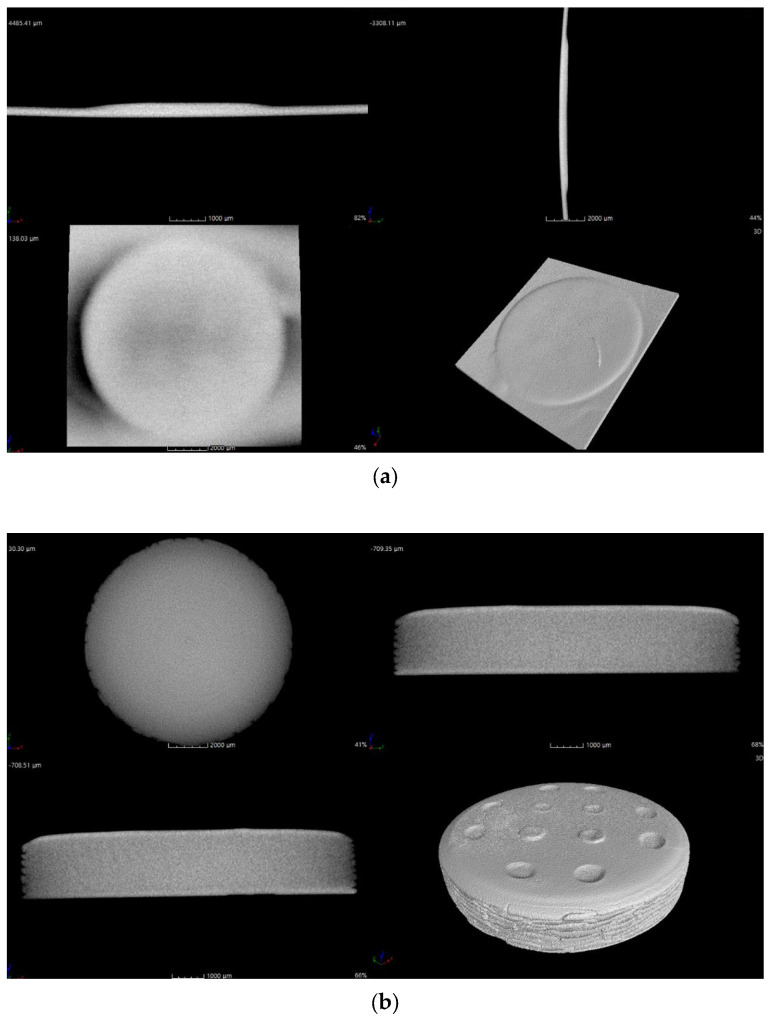
X-ray CT cross-sectional images of printed Al_2_O_3_-acrylate composites with different SCA concentrations; (**a**) 2 wt.%, (**b**) 5 wt.%, (**c**) 10 wt.%, and (**d**) 20 wt.%.

**Table 1 materials-17-05541-t001:** Fitting parameters of photocurable Al_2_O_3_ slurries with different SCA concentrations using Herschel–Bulkley model. Yield stress (τ_sr_) determined by stress ramp test is presented for comparison.

SCA Content (wt.%)	Yield Stress (τ_0_)	Consistency Index (k)	Flow Index (n)	R^2^	Yield Stress (τ_sr_)
2	3721.6	1261.3	0.819	0.9917	765
5	1436.5	811.1	0.606	0.9864	586
10	340.8	215.3	0.843	0.9966	43
20	257.6	208.6	0.973	0.9989	3

**Table 2 materials-17-05541-t002:** Cure parameters for photocurable Al_2_O_3_ slurries with various silane coupling agent concentrations. SD stands for Standard Deviation.

SCA Content (wt.%)	Exposure Time (t)	Maximum Heat Flow (mW/mg) ± SD	Curing Enthalpy (J/g) ± SD	Conversion (%) ± SD
2	1.2	3.29 ± 0.15	77 ± 5	51 ± 3
	2.4	4.12 ± 0.20	105 ± 6	70 ± 4
	4.8	5.82 ± 0.25	111 ± 7	74 ± 5
5	1.2	3.12 ± 01.0	74 ± 4	46 ± 2
	2.4	4.98 ± 0.18	117 ± 8	73 ± 6
	4.8	7.63 ± 0.30	160 ± 10	100 ± 10
10	1.2	4.06 ± 0.14	77 ± 5	45 ± 3
	2.4	6.12 ± 0.20	97 ± 6	57 ± 4
	4.8	6.25 ± 0.22	122 ± 8	71 ± 5
20	1.2	3.51 ± 0.11	62 ± 3	34 ± 2
	2.4	4.34 ± 0.16	88 ± 5	48 ± 3
	4.8	5.78 ± 0.19	111 ± 7	61 ± 4

## Data Availability

The original contributions presented in the study are included in the article, further inquiries can be directed to the corresponding authors.

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
