# Peer review of "Impact of Optimal Silane Concentration on the Rheological Properties and 3D Printing Performance of Al2O3-Acrylate Composite Slurries"

_materials, 2024, doi:10.3390/ma17225541_

Round 1

Reviewer 1 Report

Comments and Suggestions for Authors

In this study, a silane agent has been used to modify the surface of Al2O3 powder. The rheological properties and photocuring characteristics of the composite slurries have been studied, as well as the 3D printing assays that have been carried out. Based on findings, the authors confirmed that the silane group forms bonds on the Al2O3 surface, significantly impacting the stability, rheological properties, and printing quality of Al2O3-acrylate composite slurries. Finally, a concentration of 5 wt.% of silane was found to be optimal, contributing to the formation of stable and robust structures. The research design is appropriate, and the experiments were technically performed. However, there are some concerns that should be addressed.

Comments

1.      The title should be more attractive.

2.      Were there any statistical calculations performed on the experiments through the manuscript? If so, they should be discussed.

3.      Graphs in Figures 2 and 8 must be compared according to p levels. What is the significance of the difference between the results according to the p level? In this case, the authors should use an ANOVA test.

4.      The authors should justify the choice of the “coaxial cylinder” geometry against the cone-plate geometry. Furthermore, what is the diameter of such geometry used?

5.      Based on the LVR assays displayed in Figure 4, the authors should precise with the value of fixed stress (Pa) that has been used to carry out the frequency sweep under LVR (Figure 5).

6.      The legend of figure 5 should be more informative, including temperature and shear stress value, and precise in that LVR is applied.

7.      The legend of Figure 4 should be more informative, including temperature and applied fixed frequency.

8.      The appropriately rheological model for Figure 6B is Carreau ou Cross with an interesting Newtonian viscosity (eta zero at low shear rate). The authors could then predict different parameters thanks to the applied model, such as relaxation time, eta zero at low shear rate, flow index, etc.

9.      The results and discussion concerning the rheological part (Lines 184-229) should be supported with references, and the findings should be compared to the literature. 3

10.   Values in Table 1 should include the SD if the presented results are averages of various measurements (means +/- SD).

Author Response

Dear Reviewer

Thank you for reviewing our manuscript and for providing us with your comments. Our point-by-point responses to your comments are set out below. We have made every effort to fully address all of your concerns and hope the manuscript is now acceptable for publication in the Materials. If you require any further information, please do not hesitate to contact me.

Comment 1.The title should be more attractive.

(Response) Suggested title is as follows: Impact of Optimal MPTMS Concentration on the Rheological Properties and 3D Printing Performance of Al2O3-Acrylate Composite Slurries

Comment 2. Were there any statistical calculations performed on the experiments through the manuscript? If so, they should be discussed.

(Response) As suggested, statistical analysis was conducted on the data shown in Figures 2 and 8, and the results have been summarized in the main text.

Comment 3. Graphs in Figures 2 and 8 must be compared according to p levels. What is the significance of the difference between the results according to the p level? In this case, the authors should use an ANOVA test.

(Response) The change in adsorption amount according to the SCA addition levels was analyzed using ANOVA to determine if the results were statistically significant. The analysis yielded a p-value of 0, indicating that the change in adsorption amount due to varying SCA addition levels is meaningful.

From my understanding, when the variable x is continuous, regression analysis is preferred over ANOVA. In the case of performing regression analysis on the TSI values of slurries with time for different SCA concentrations, the p-values are all below 0.05.

We have added the above content to the text right above Figures 2 and 8.

Comment 4. The authors should justify the choice of the “coaxial cylinder” geometry against the cone-plate geometry. Furthermore, what is the diameter of such geometry used?

(Response) We have added the following explanation to the Materials and Methods. The source was also referenced. “The cup-and-bob configuration is beneficial for low-viscosity fluids because it increases the total surface area and, consequently, the viscous drag on the rotating inner cylinder, thereby enhancing the accuracy of the measurements.”

The diameter of cylinder used in this study was 16 mm. This information was added in the line 97.

Comment 5. Based on the LVR assays displayed in Figure 4, the authors should precise with the value of fixed stress (Pa) that has been used to carry out the frequency sweep under LVR (Figure 5).

(Response) Based on the LVR analysis results in Figure 4, the value of the fixed stress used during the frequency sweep measurement in Figure 5 was 100 Pa. Additionally, all rheometer measurements were conducted at room temperature (25°C).

Comment 6. The legend of figure 5 should be more informative, including temperature and shear stress value, and precise in that LVR is applied.

(Response) The legend of Figure 5 has been revised as follows: Figure 5. Results of frequency sweep test for the Al2O3 slurries with different SCA concentrations; (a) 2 wt.%, (b) 5 wt.%, (c) 10 wt.%, and (d) 20 wt.%. All the measurements were conducted at 25 ℃ and a shear stress of 100 Pa.

Comment 7. The legend of Figure 4 should be more informative, including temperature and applied fixed frequency.

(Response) The legend of Figure 4 has been revised as follows: Figure 4. Results of amplitude sweep test for the Al2O3 slurries with different SCA concentrations; (a) 2 wt.%, (b) 5 wt.%, (c) 10 wt.%, and (d) 20 wt.%. All the measurements were conducted at 25 ℃ and a frequency of 10 rad/s.

Comment 8. The appropriately rheological model for Figure 6B is Carreau ou Cross with an interesting Newtonian viscosity (eta zero at low shear rate). The authors could then predict different parameters thanks to the applied model, such as relaxation time, eta zero at low shear rate, flow index, etc.

(Response) The rheological behavior of the slurries shown in Figure 6(b) was fitted using the Carreau-Yasuda model, with the fitting parameters summarized in Table 1. The zero shear viscosity decreased with the addition of SCA. The time constant was found to be independent of the amount of SCA added, with the 10 wt.% SCA slurry exhibiting the lowest value. The transition control factor and power index were consistent across all slurries.

Comment 9. The results and discussion concerning the rheological part (Lines 184-229) should be supported with references, and the findings should be compared to the literature.

(Response) The rheological part was revised using the results from the references and a rheology model (Carreau-Yasuda Model).

Comment 10. Values in Table 1 should include the SD if the presented results are averages of various measurements (means +/- SD)

(Response) We have added the standard deviation values in the Table 2.

Reviewer 2 Report

Comments and Suggestions for Authors

The manuscript presents the rheological properties, photocuring characteristics, and 3D printing performance of MPTMS surface modifier.

In the last paragraph of the introduction section, I recommend that the authors should clarify the main objective because it looks like two objectives in the paragraph.

In line 74, can the authors add any reference about the concentration indicated? Also, I recommend that the authors should include a reference on the preparation of Al2O3 acrylate composite slurry.  

The authors should include the measure conditions of oscillation sweep tests because they omitted details in this context.

In general, I recommend including a reference in the methodologies presented in its manuscript.  

The authors omitted the constant strain value in the frequency sweep measurements.

I recommend that the authors should include a statistical analysis to verify and validate the effects of the concentrations evaluated.

I recommend that the authors improve Figure 1 with color lines for each treatment because the analysis is complicated in this way.

I recommend that the authors present all treatments in one plot.

 In Figure 6b, I recommend that the authors calculate the theological parameters fitting a mathematical model of the flow curves.

I recommend that the authors improve Figure 7 with color lines for each treatment because the analysis is complicated in this way.

Table 1 should include standard deviation and statistical analysis.

Author Response

Dear Reviewer

Thank you for reviewing our manuscript and for providing us with your comments. Our point-by-point responses to your comments are set out below. We have made every effort to fully address all of your concerns and hope the manuscript is now acceptable for publication in the Materials. If you require any further information, please do not hesitate to contact me.

Comment 1. The manuscript presents the rheological properties, photocuring characteristics, and 3D printing performance of MPTMS surface modifier.

In the last paragraph of the introduction section, I recommend that the authors should clarify the main objective because it looks like two objectives in the paragraph.

(Response) To clarify the purpose of the study, the final paragraph has been revised as follows: “In this study, 3-trimethoxy-silylpropane-1-thiol (MPTMS) was used as a surface modifier for Al2O3 powder to systematically analyze the effect of MPTMS concentration on the rheological properties, photocuring characteristics, and 3D printing performance of the slurry. MPTMS plays a crucial role in controlling particle interactions by forming silane bonds on the Al2O3 surface, thereby adjusting the slurry's stability and curing rate. The primary objective is to identify the optimal MPTMS concentration that maximizes the physical properties of the Al2O3 slurry and enhances 3D printing performance. This research will provide fundamental data crucial for the production of high-quality ceramic components.”

Comment 2. In line 74, can the authors add any reference about the concentration indicated? Also, I recommend that the authors should include a reference on the preparation of Al2Oacrylate composite slurry.

(Response) References are added as recommended.

Comment 3. The authors should include the measure conditions of oscillation sweep tests because they omitted details in this context.

(Response) Amplitude sweep test was conducted at 25 ℃ and a frequency of 10 rad/s. This was explained in the figure caption.

Comment 4. In general, I recommend including a reference in the methodologies presented in its manuscript.

(Response) References are added as recommended.

Comment 5. The authors omitted the constant strain value in the frequency sweep measurements.

(Response) Based on the LVR analysis results in Figure 4, the value of the fixed stress used during the frequency sweep measurement in Figure 5 was 100 Pa. Additionally, all rheometer measurements were conducted at room temperature (25°C).

Comment 6. I recommend that the authors should include a statistical analysis to verify and validate the effects of the concentrations evaluated.

(Response) As suggested by the reviewer, statistical analysis was conducted on the data shown in Figures 2 and 8, and the results have been summarized in the main text.

The change in adsorption amount according to the SCA addition levels was analyzed using ANOVA to determine if the results were statistically significant. The analysis yielded a p-value of 0, indicating that the change in adsorption amount due to varying SCA addition levels is meaningful.

When the variable x is continuous, regression analysis is preferred over ANOVA. In the case of performing regression analysis on the TSI values of slurries with time for different SCA concentrations, the p-values are all below 0.05.

Comment 7. I recommend that the authors improve Figure 1 with color lines for each treatment because the analysis is complicated in this way.

(Response) Figure 1 was improved with color lines.

Comment 8. I recommend that the authors present all treatments in one plot.

(Response) To enhance data readability, we aimed to present the results measured using the same analytical method in a single figure, excluding separately drawn figures. This approach was taken to analyze the effects of concentration on the slurry and photocuring behavior. If the reviewer indicates any additional figures that need to be revised, we will make the necessary adjustments.

Comment 9. In Figure 6b, I recommend that the authors calculate the theological parameters fitting a mathematical model of the flow curves.

(Response) The rheological behavior of the slurries shown in Figure 6(b) was fitted using the Carreau-Yasuda model, with the fitting parameters summarized in Table 1. The zero shear viscosity decreased with the addition of SCA. The time constant was found to be independent of the amount of SCA added, with the 10 wt.% SCA slurry exhibiting the lowest value. The transition control factor and power index were consistent across all slurries.

Comment 10. I recommend that the authors improve Figure 7 with color lines for each treatment because the analysis is complicated in this way.

(Response) Figure 7 was improved with color lines.

Comment 11. Table 1 should include standard deviation and statistical analysis.

(Response) We have added the standard deviation values in the table.

Reviewer 3 Report

Comments and Suggestions for Authors

This work investigates the effect of a new coupling agent (MPTMS) concentration in a ceramic-polymer composite. The rheological properties, dispersion stability and 3-D printing behaviour of the material in the paste form was studied.

The Abstract is clear and informative, containing the main achievements of the work.

The Introduction section is short and based mainly on previous reviews. It is not bad, however some works closer to the subject treated by this article’s authors should be mentioned and described in a few sentences or phrases, to frame this work in the state-of-the-art.

The Materials and Methods section is very well organized and described so the experiment’s replication become possible.

The Results and Discussion section contains the following:

-Adsorption of coupling agent (abbreviation SCA) on Al2O3 was studied by thermogravimetric analysis, rising the temperature from environmental to 800 oC. The stages of adsorption-desorption and eventual decomposition were identified on the thermogravimetric curve. Also, the effect of SCA concentration on the weight loss of the material led to interesting discussion linked to the change in the material composition.

-The FT-IR analysis served to identify the changes in the material composition due to the reactions in which the components entered.

-The rheological study of the photocurable slurries revealed that all slurries exhibited elastic deformation specific to gels. The linear viscoelastic range (LVR) decreased at concentration of SCA> 10% in slurry, thus indication the weakening in the slurry resistance to deformation.

-The dispersibility and sedimentation of photocurable slurries was characterized by Turbiscan Stability Index showing that 5% SCA in the slurry is the best from dispersibility point of view, thusresulting a more homogeneous material prone to further processing.

-The photocuring process was studied and the material containing 5%SCA proved the highest heat flow and the highest conversion. For an exposure of 4.8 seconds, the conversion reaches 100% after 2 minutes curing.

-The 3-D printing behaviour was studied versus SCA content. The slurry containing 5% had the only good behaviour by forming a strong network during photopolymerization. The paste containing 2% SCA failed to adhere to the build platform since at higher content (10% or 20%) led to cracks in the material observed at micro-CT.

The Discussion section could be improved by comparison with similar materials in the literature.

The Conclusions section reviews the main findings of the work. A view to the continuation of this work would be beneficial (new directions of future work, industrial applications).

Author Response

Dear Reviewer

Thank you for reviewing our manuscript and for providing us with your comments. Our point-by-point responses to your comments are set out below. We have made every effort to fully address all of your concerns and hope the manuscript is now acceptable for publication in the Materials. If you require any further information, please do not hesitate to contact me.

Comment 1. This work investigates the effect of a new coupling agent (MPTMS) concentration in a ceramic-polymer composite. The rheological properties, dispersion stability and 3-D printing behaviour of the material in the paste form was studied.

The Abstract is clear and informative, containing the main achievements of the work.

(Response) Thank you for your thoughtful review and comments.

Comment 2. The Introduction section is short and based mainly on previous reviews. It is not bad, however some works closer to the subject treated by this article’s authors should be mentioned and described in a few sentences or phrases, to frame this work in the state-of-the-art.

(Response) As recommended, part of introduction has been revised as follows with references: “Many studies have been conducted to improve the properties of 3D printing processes and photocurable structures using various ceramic powders modified with SCA [6-9]. The powders modified with SCA exhibit excellent dispersibility, improving the uniformity and fluidity of the slurry, which in turn enhances the mechanical properties of the printed ceramic polymer composites. However, most studies have focused on comparing and evaluating different types of SCA or conducting experiments with lower concentrations of additives for practical manufacturing applications. Some studies have concentrated on analyzing the chemical bonding formed when powders modified with SCA are mixed with monomers or oligomers at each stage of the experiment [13-15]. While research has been reported on a wide range of SCA concentrations to increase the solid content of the slurry, these studies have mostly focused on changes in slurry viscosity, without analyzing the effects on photocuring characteristics [16]. Some studies have reported changes in material properties after curing, but systematic research on the changes in material characteristics at each process stage and the correlation between those characteristics and the properties after curing is still lacking [17-19].”

Comment 3. The Materials and Methods section is very well organized and described so the experiment’s replication become possible.

(Response) Thank you for your thoughtful review and comments.

Comment 4. The Results and Discussion section contains the following:

-Adsorption of coupling agent (abbreviation SCA) on Al2O3 was studied by thermogravimetric analysis, rising the temperature from environmental to 800 oC. The stages of adsorption-desorption and eventual decomposition were identified on the thermogravimetric curve. Also, the effect of SCA concentration on the weight loss of the material led to interesting discussion linked to the change in the material composition.

-The FT-IR analysis served to identify the changes in the material composition due to the reactions in which the components entered

-The rheological study of the photocurable slurries revealed that all slurries exhibited elastic deformation specific to gels. The linear viscoelastic range (LVR) decreased at concentration of SCA> 10% in slurry, thus indication the weakening in the slurry resistance to deformation.

-The dispersibility and sedimentation of photocurable slurries was characterized by Turbiscan Stability Index showing that 5% SCA in the slurry is the best from dispersibility point of view, thus resulting a more homogeneous material prone to further processing.

-The photocuring process was studied and the material containing 5%SCA proved the highest heat flow and the highest conversion. For an exposure of 4.8 seconds, the conversion reaches 100% after 2 minutes curing.

-The 3-D printing behaviour was studied versus SCA content. The slurry containing 5% had the only good behaviour by forming a strong network during photopolymerization. The paste containing 2% SCA failed to adhere to the build platform since at higher content (10% or 20%) led to cracks in the material observed at micro-CT.

The Discussion section could be improved by comparison with similar materials in the literature.

(Response) Discussion part was modified with statistical analysis as follows:

Statistical analysis was conducted on the data shown in Figures 2 and 8, and the results have been summarized in the main text.

The rheological part was revised using the results from the references and a rheology model (Carreau-Yasuda Model).

Commnet 5. The Conclusions section reviews the main findings of the work. A view to the continuation of this work would be beneficial (new directions of future work, industrial applications).

(Response) Future work was added in the conclusion part as follows: “This study presents the technical guidance for preparing photocurable slurry required for the manufacturing of ceramic-polymer composites using a 3D printing process. Future research will aim to supplement the analysis of property changes during long-term storage of the slurry and the mechanical properties of the composite after curing, in order to establish the technical indicators necessary for practical application in manufacturing settings.”

Round 2

Reviewer 1 Report

Comments and Suggestions for Authors

The authors addressed all my concerns. However, the minor issues listed below should be considered:

1.      The term “MPTMS” in the title should be replaced with “Silane”.

2.      The protocol used for the statistical analysis must be presented as a sub-section (i.e., 2.5. statistical analysis) in the M&M section as

3.      Line 154: the p-value should be corrected.

4.      Legends of figures 2 and 8 should include the significance of the difference between the results according to the p level.

Author Response

Dear Reviewer

Thank you for reviewing our manuscript and for providing us with your comments. Our point-by-point responses to your comments are set out below. We have made every effort to fully address all of your concerns and hope the manuscript is now acceptable for publication in the Materials. If you require any further information, please do not hesitate to contact me.

Comment 1. The term “MPTMS” in the title should be replaced with “Silane”.

(Response) Title was modified as follows: “Impact of Optimal Silane Concentration on the Rheological Properties and 3D Printing Performance of Al2O3-Acrylate Composite Slurries”

Comment 2. The protocol used for the statistical analysis must be presented as a sub-section (i.e., 2.5. statistical analysis) in the M&M section as

(Response) Statistical analysis was added in the M&M as follows:

2.5. Statistical Analysis 

The significance test between each treatment group for the experimental results was conducted using statistical analysis with Minitab (Minitab, INC., USA) and verified at a significance level of p < .05. The change in adsorption amount according to the SCA addition levels was analyzed using one-way ANOVA to determine if the results were statistically significant. TSI values of slurries with time for different SCA concentrations were analyzed using regression analysis.

Comment 3. Line 154: the p-value should be corrected.

(Response) p-value was corrected as follows: p <.001

Comment 4. Legends of figures 2 and 8 should include the significance of the difference between the results according to the p level.

(Response) The confidence level was added in the figure legends.

Reviewer 2 Report

Comments and Suggestions for Authors

The manuscript is accepted.

Author Response

Thank you for reviewing the article.